# Manganese Deficiency Suppresses Growth and Photosynthetic Processes but Causes an Increase in the Expression of Photosynthetic Genes in Scots Pine Seedlings

**DOI:** 10.3390/cells11233814

**Published:** 2022-11-28

**Authors:** Yury V. Ivanov, Pavel P. Pashkovskiy, Alexandra I. Ivanova, Alexander V. Kartashov, Vladimir V. Kuznetsov

**Affiliations:** K.A. Timiryazev Institute of Plant Physiology, Russian Academy of Sciences, Botanicheskaya Street 35, 127276 Moscow, Russia

**Keywords:** *Pinus sylvestris*, growth retardation, ion homeostasis, photosystem II, photosynthetic genes

## Abstract

Manganese deficiency is a serious plant nutritional disorder, resulting in the loss of crop productivity in many parts of the world. Despite the progress made in the study of angiosperms, the demand for Mn in gymnosperms and the physiological responses to Mn deficiency remain unexplored. We studied the influence of Mn deficiency for 24 weeks on *Pinus sylvestris* L. seedling growth, ion homeostasis, pigment contents, lipid peroxidation, chlorophyll fluorescence indices and the transcript levels of photosynthetic genes and genes involved in chlorophyll biosynthesis. It was shown that Mn-deficient plants demonstrated suppressed growth when the Mn content in the needles decreased below 0.34 µmol/g DW. The contents of photosynthetic pigments decreased when the Mn content in the needles reached 0.10 µmol/g DW. Mn deficiency *per se* did not lead to a decrease in the nutrient content in the organs of seedlings. Photoinhibition of PSII was observed in Mn-deficient plants, although this was not accompanied by the development of oxidative stress. Mn-deficient plants had an increased transcript abundance of genes (*psbO*, *psbP*, *psbQ*, *psbA* and *psbC*), encoding proteins directly associated with the Mn cluster also as other proteins involved in photosynthesis, whose activities do not depend on Mn directly. Furthermore, the transcript levels of the genes encoding the large subunit of Rubisco, light-dependent NADPH-protochlorophyllide oxidoreductase and subunits of light-independent protochlorophyllide reductase were also increased in Mn-deficient plants.

## 1. Introduction

Manganese (Mn) is an important nutrient for multiple ecosystem processes, ranging from photosynthesis to degradation of lignin in litter and litter residues [1]. In plants, Mn is one of the 17 essential elements critical for growth and reproduction [2]. Plants have an essential dependence on Mn owing to its indispensable role in the oxygen-evolving complex (OEC) of photosystem II (PSII), which catalyzes the oxidation of water to protons and molecular oxygen [3,4]. Therefore, insufficient Mn supply leads to decreased oxygen evolution and, hence, to lower rates of photosynthesis and decreased plant growth [5,6].

Mn shows the greatest variation among some annual and perennial species, indicating that leaf tissue is capable of buffering fluctuations in the root uptake of Mn. This is probably an evolutionary adaptation because in plants growing in soil, fluctuations in the uptake of Mn may be stronger than those of other nutrients [7]. For a long time, the accepted dogma among plant biologists has been that the physiological requirement for Mn by living cells is low and that Mn uptake exceeds the requirement [2]. However, in natural and agricultural settings, Mn availability can be a seriously limiting factor for plant growth [8].

Although Mn is one of the most abundant trace elements in the lithosphere, its contents in soils are highly diverse [9]. Mn bioavailability in soils is negatively correlated with pH and redox potential. Plant uptake of Mn is essentially independent of pH from pH 5.5 to 8.0 and increases rapidly with increased acidity below pH 5.5 [10]. Alkaline and oxidative (well-aerated) conditions favor the formation of Mn oxides that are unavailable to plants [6]. Thus, Mn deficiency is abundant in plants growing in soils derived from parent material inherently low in Mn and in highly leached soils. It is also common in soils of high pH containing free carbonates, particularly when combined with high organic matter content [5]. Podzols and deep sands are known to be associated with Mn deficiency worldwide. In general Mn concentrations are lowest between 0.1 m and 0.6 m soil depths [11,12].

Mn budgets (input/output ratios) in soils of various ecosystems indicated a predominance of leaching processes over atmospheric input. In pine, spruce and birch forests, leaching from soil profiles accounts for 360–11,000 g ha^−1^ yr^−1^, whereas in other ecosystems some accumulation (91–191 g ha^−1^ yr^−1^) of Mn was observed [9]. Several studies have indicated that Mn is a relatively immobile element in pine ecosystems. Harvesting of stemwood with bark removes a greater proportion of the Mn in the system than of other micronutrients [13]. Estimates of the proportion of Mn removed vary from 52–63% [11] to 76% [14] and increase with stand age. Thus, logging is an additional factor of available Mn depletion in forest soils in which reserves are not supplied by fertilization, as in agricultural lands.

Coniferous species are well known as Mn accumulators and high foliar concentrations have been recorded with little apparent influence on growth [11]. Thus, newly shed leaf litter of Scots pine (*Pinus sylvestris* L.) had an Mn content ten-fold higher than that of the leaf litter of grey alder (*Alnus incana* L.) on the same site [1]. However, there is a large variability in leaf Mn content both between different genera and within the same genus. For example, the average Mn content for Norway spruce (*Picea abies* (L.) H. Karst) at least two-fold higher than that of Scots pine [1,15]. According to the Mn foliar content some species from the genus *Pinus*, could be ranked in the following orders: *P. resinosa* > *P. virginiana* > *P. banksiana* = *P. radiata* > *P. elliotti* > *P. taeda* [13] and *P. contorta* > *P. sylvestris* > *P. densiflora* > *P. thunbergiana* > *P. nigra* [1,16]. Unfortunately, there are no data to combine these species into one continuous order. Nevertheless, even for one species (specifically Scots pine) Mn contents varied considerably within a given stand for 17 consecutive years (range factor of 5.0) [1]. Moreover, variation (up to 76.4%) in Mn content was noted for needles of Scots pine seedlings grown in hydroculture, while available Mn concentrations in the nutrient solutions were the same [17,18].

Mn deficiency strongly affects photosynthesis; however, visual symptoms of chlorosis, reflecting photochemical disturbances, are only visible when plant growth is severely depressed [2,5]. For the same reason, the magnitude of Mn deficiency is expected to constitute a much larger problem than would be expected from visual observations in the field [8]. At the onset of Mn deficiency, the symptom stratification is confined to the newly emerged leaves as a result of the low phloem mobility of Mn that prevents remobilization of Mn from older to younger leaves [8].

Despite differences in efficiency among plant species, the critical deficiency contents of Mn in crops are similar, varying between 10–20 µg/g DW (0.18–0.36 µmol/g DW) in fully expanded leaves [5]. However, for most coniferous species there is no definitive information on the critical level of Mn in foliage [19]. The optimal range of mass-based Mn content in Scots pine was estimated at 1.27–7.28 µmol/g for one-year-old needles [20]. Nevertheless, Kavvadias and Miller [19] found clear indications that the growth of Scots pine seedlings is suboptimal at foliar contents below 1.46 µmol/g, while Brække and Salih [21] estimated strong Mn deficiency in current-year-needles at 0.18 µmol/g. Grey [11] indicated that foliar levels below 0.55 µmol/g may be associated with reduced diameter growth in *P. radiata*, and Morrison and Armson [22] found that foliar content of Mn at 0.18 µmol/g suppressed growth of seedlings of *P. banksiana* Lamb.

In addition to its crucial role in photosynthesis, Mn is an important cofactor of enzymes involved in isoprenoid biosynthesis [2] that are responsible for pathogen defense and adaptation to other stressors. Thus, the stability and productivity of pine stands may depend on Mn availability. Previously, we found that in response to different stressors, such as heavy metals [17,18,23], soil alkalization [24] or water shortages [15], the Mn content in the organs of Scots pine changed much more strongly than other nutrients. In most cases, these changes were associated with a drastic decrease in the Mn content in plant organs. However, the physiological consequences of this phenomenon are still not clear. Moreover, the regulation of gene expression under Mn deficiency has not been sufficiently studied. In particular, it is not clear how Mn deficiency affects the transcript levels of genes encoding Mn-dependent enzymes and other enzymes not directly related to Mn.

Therefore, the aims of our research were (1) to identify the minimal Mn content in the organs of Scots pine seedlings, initiating disturbances of growth during Mn deficiency of long duration (up to 168 days) after seed germination; (2) to investigate the response of growth, ion homeostasis and content of photosynthetic pigments; (3) to investigate photosynthetic processes and lipid peroxidation in the needles; and (4) to explore the possible changes in the transcript levels of photosynthetic genes, especially the genes encoding OEC, and genes involved in chlorophyll biosynthesis.

## 2. Materials and Methods

### 2.1. Experimental Design

Seeds of Scots pine (*P. sylvestris* L.) were collected in the Bryansk region (Bryansk, Russia) from high-productive pine stands in complex forest types. The seedlings were cultivated in hydroculture, as previously described [17], with actual concentrations of Mn (26.3 nM [Mn-deficient] and 5.2 [control] μM). The nutrient solutions were constantly aerated and renewed once a week. The seedlings were cultivated until the age of 24 weeks in a growth chamber that provided a constant air temperature of 24 ± 2 °C and a photoperiod of 16 h under fluorescent lighting (L36W/765, JSC OSRAM, Russia, 130 ± 15 μmol m^−2^ s^−1^).

The plants were collected at the 4th, 6th, 12th, 19th and 24th weeks after seed germination. Of the plants collected at the 4th and 6th weeks, 3–5 seedlings were grouped into composite samples and dissected into their root system, hypocotyl, cotyledons, and needles for the determination of the Mn content and calculation of Mn amount per seedling. Of the plants collected at the same time points (from the 12th week onwards), 1 or 2 seedlings were grouped into composite samples and dissected into their root system and needles for the determination of the Mn and other essential elements, fresh weight and water contents. At the 6th, 19th and 24th weeks of the experiment, the roots and needles of plants were collected, fixed in liquid nitrogen, and stored at −70 °C until the biochemical analyses. At the 24th week of the experiment, the upper needles of plants were collected for the determination of the nutrient contents. Each composite sample was treated as a biological replicate.

### 2.2. Determining the Fresh Weight and Water Content

The fresh weights of the seedlings, root system and needles were determined with an accuracy of 1 mg using an analytical balance (Scout Pro SPU123, Ohaus Corporation, Parsippany, NJ, USA). The dry weight was determined using an analytical balance (AB54-S, Mettler Toledo, Switzerland) with an accuracy of 0.1 mg after drying the samples to a constant weight at 70 °C. The water content of each organ is expressed as a percentage of its fresh weight [18].

### 2.3. Determining the Morphometric Parameters

To determine the linear dimensions of 4- and 6-week-old seedlings, they were laid out individually on the glass of an Epson Perfection V500 Photo flatbed scanner (Epson, Japan) and scanned at a resolution of 800 dpi. MapInfo Professional v. 9.5 software (Pitney Bowes Software, Stamford, CT, USA) was used to measure the lengths of the seedling organs (primary root, hypocotyl, cotyledons, needles) and the distance from the tip of the main root to the first lateral root to an accuracy of 0.01 mm and to count the number of first-order lateral roots and needles of the seedlings [17].

### 2.4. Determining the Contents of Essential Elements

To determine the contents of Mn, Mg, P, K, Ca, Fe, Zn and Cu ions in the organs, the seedling roots were washed in a 20 mM aqueous solution of Na_2_-EDTA to remove the ions adsorbed on the surface. Next, the roots were thoroughly rinsed in distilled water and blotted on filter paper. Thereafter, the seedlings were dissected into the roots and needles and dried until reaching a constant weight. Seed Mn content was determined in full seeds prepared as described above. The samples were then digested in solutions of concentrated HNO_3_ and HClO_4_ (2:1 (*v*/*v*)), after which the nutrient contents (except P) were determined by atomic absorption spectrometry [17,18]. Phosphorous was determined spectrophotometrically using a molybdenum blue reaction [25] as described in Ivanov et al. [15].

### 2.5. Determination of Photosynthetic Pigments

The chlorophyll *a* (Chl *a*), *b* (Chl *b*) and carotenoid (Car) contents were determined using the Lichtenthaler method [26]. The samples were triturated with 80% acetone in the dark. The absorbance of the centrifuged samples was measured with a Genesys 10 UV–Vis spectrophotometer (Thermo Fisher Scientific, Waltham, MA, USA) at wavelengths of 470, 646, and 663 nm. The content of the photosynthetic pigments was determined using the Lichtenthaler formulas [26]:Chl *a* = 12.25 × A_663_ − 2.79 × A_646_,(1)
Chl *b* = 21.50 × A_646_ − 5.10×A_663_,(2)
Car = (1000 × A_470_ − 1.82 × Chl *a* − 85.02 × Chl *b*)/198,(3)

### 2.6. Evaluating the Level of Lipid Peroxidation

The contents of malondialdehyde (MDA) and 4-hydroxy-2-nonenal (4-HNE) were determined spectrophotometrically, with maximum optical absorption at 586 nm, by measuring the product that formed during the reaction using the selective reagent 1-methyl-2-phenylindole (Aldrich, CAS Number 3558-24-5) in accordance with Gérard-Monnier et al. [27]. To construct the calibration curve, 1,1,3,3-tetraethoxypropane was used [17].

### 2.7. Determination of Chlorophyll Fluorescence

The fluorescence induction curves were measured with a MINI-PAM-II/B fluorometer (Walz, Effeltrich, Germany). After a pulse of saturating light, the leaves of plants adapted to 30 min in the dark were kept in the dark for one minute and then they were exposed to actinic light for 5 min, which was followed by saturating light pulses during which the parameters were measured. Blue LEDs (470 nm) were used to provide the measuring light (0.05 µmol photons m^−2^ s^−1^), actinic light (125 µmol photons m^−2^ s^−1^) and saturating pulses (470 nm, 5000 µmol photons m^−2^ s^−1^ and 800 ms duration). The parameter calculations on the basis of fluorescence data were performed using WinControl-3 v.3.32 software (Walz, Effeltrich, Germany), and the formulas were taken from Schreiber [28]. The values for F_0_, F_v_, F_m_, F_m_’ and F_0_’, as well as the PSII maximum (F_v_/F_m_) and effective Y(II) (F_m_’ − F_t_)/F_m_’ photochemical quantum yields and non-photochemical quenching (NPQ) (F_m_/F_m_’ − 1) were determined. F_m_ and F_m_’ are the maximum Chl fluorescence levels under dark- and light-adapted conditions, respectively. F_v_ is the photoinduced change in fluorescence, and F_t_ is the level of fluorescence before a saturation impulse is applied. F_0_ is the initial Chl fluorescence level. The quenching parameters were also determined: NPQ—non-photochemical fluorescence quenching; Y(NO)—quantum yield of non-regulated non-photochemical energy dissipation in PSII and Y(NPQ)—quantum yield of regulated non-photochemical energy dissipation in PSII, Y(NO) + Y(NPQ) + Y(II) = 1.

### 2.8. RNA Extraction and Quantitative RT-PCR

RNA isolation was performed according to the method of Kolosova et al. [29], with some modifications described in Pashkovskiy et al. [30]. cDNA synthesis and qRT-PCR analysis of gene expression patterns were performed according to Pashkovskiy et al. [31]. The list of gene-specific primers is given in Appendix A. The transcript levels were normalized to the expression of the *Actin1* gene. The relative changes in the expression level of the genes at all time points and in all variants in comparison to the average expression level of the control plants at the 6th week of the experiment were then calculated.

### 2.9. Statistical Analysis

The number of biological replicates in the determination of fresh biomass of the seedlings and the seedling organs ranged from 15 to 40. The number of biological replicates to determine the seedling organ water content ranged from 6 to 12. The number of biological replicates assessed in the determination of the morphometric parameters of 4- and 6-week-old seedlings was 80. Six biological replicates were performed to determine the nutrient contents, chlorophyll fluorescence and lipid peroxidation parameters, pigment content and transcript levels of genes.

The data were statistically analyzed using SigmaPlot 12.3 (Systat Software Inc., San Jose, CA, USA). The values presented in the tables and figures are the arithmetic means ± standard errors. Pairwise comparisons of the means were performed using Student’s *t* test for normally distributed data or the Mann–Whitney rank sum test when the *t* test was not applicable. Asterisks (*, for Student’s *t* test) or multiplication signs (×, for Mann–Whitney rank sum test) denote significant differences at *p* < 0.05 between the control and experimental variants at each time point.

## 3. Results

### 3.1. The Mn Content in the Nutrient Solutions, Seeds and Seedling Organs

Atomic absorption spectrometry data indicate the identity between the nominal (5.0 μM) [17] and measured (5.21 ± 0.33 μM) Mn concentrations in nutrient solution for control plants (hereinafter referred to as “control”). Due to admixtures of Mn in the mineral salts used to prepare the nutrient solution without addition of Mn (hereinafter referred to as “Mn-deficient”), its concentration reached 26.3 ± 2.2 nM. Thus, the differences in Mn concentrations between the control and Mn-deficient nutrient solutions reached 200 times.

For the seeds used in the experiment, the Mn content was 42.4 ± 1.6 nanomoles per individual seed.

The Mn content in the roots of control plants varied in the range of 0.57–3.17 µmol/g DW (average 1.45 µmol/g) during the experiment, with a maximum at the 4th and 6th weeks of the experiment (Figure 1A). The Mn content in the roots of Mn-deficient plants varied within a smaller range, 0.04–0.18 µmol/g DW, and was maintained at an average level of 0.10 µmol/g until the end of the experiment (Figure 1A).

The Mn content in the needles of control plants stabilized after the 4th week of the experiment and averaged 6.08 µmol/g (excluding young needles). Unlike roots, a progressive decrease in the Mn content in the needles of Mn-deficient plants was observed during the experiment. The young needles of Mn-deficient plants at the end of the experiment contained 102 times less Mn than those in the control (Figure 1B).

At the 4th week, the Mn amount per seedling in Mn-deficient plants was 52.6% lower than that in the control. Within two weeks, the Mn amount per seedling in control plants increased two-fold, while in Mn-deficient plants, it increased by 30.4% only (Table 1).

### 3.2. Growth Disturbances and Ion Homeostasis under Mn Deficiency

At the 6th week of the experiment, we observed accelerated growth of Mn-deficient plants compared to the control. In particular, seedling fresh weight (FW) exceeded the control by 17.1%—root weight by 30.3% and needle weight by 15.9% (Figure 2A,B). However, starting from the 12th week, the development of Mn-deficient seedlings significantly decreased compared with the control by an average of 39.6% until the end of the experiment. The root and needle FWs of Mn-deficient plants were 40.2% and 41.1%, respectively, smaller than those in the control. Root growth was suppressed more strongly than needle growth up to the 19th week of the experiment; however, at the 24th week inhibition of needle growth prevailed over inhibition of root growth (Figure 2B).

The Mn-deficient plants showed no difference in water content in the roots compared to the control throughout the experiment (Figure 2C). In contrast, in the needles of Mn-deficient plants, starting from the 19th week, an increase in the water content was noted, reaching a maximum at the 24th week in the upper needles of 2.9% (as absolute value) more than in the control (Figure 2D).

A detailed analysis of the morphometric parameters of seedlings at the initial stages of the experiment indicates a more intensive root growth in length in Mn-deficient plants. At the 6th week of the experiment the length of the main root of Mn-deficient plants exceeded the control by 9.9%, and the distance from the tip of the main root to the first lateral root exceeded the control by 6.2%. At the same time, we did not observe differences in the number of first-order lateral roots (Table 2). The above-ground part of the seedlings was characterized by the absence of differences in the length and weight of the hypocotyl and the number of needles. However, the average length of the Mn-deficient needles exceeded that of the control by 4.8%.

We did not observe a decrease in the content of the studied mineral nutrients in the organs of seedlings during the development of Mn deficiency (Figure 3, Appendix A). In contrast, the contents of a number of mineral nutrients, namely, K, Mg, P and Fe, were increased in the needles of Mn-deficient plants at the end of the experiment by 49.8%, 27.7%, 13.3% and 30.4%, respectively (Figure 3B, Appendix A). At the 24th week of the experiment, the K and Zn contents in the roots of Mn-deficient plants were 33.5% and 2.1-fold higher, respectively, than those in the control (Figure 3A, Appendix A). The increased contents of Zn (on average by 47.5%) in the roots, as well as P in the roots (by 23.1%) and in the needles (by 47.6%) throughout the experiment were characteristic of Mn-deficient plants (Appendix A).

### 3.3. Contents of Photosynthetic Pigments and Lipid Peroxidation Products

The content of photosynthetic pigments in the needles of the control and Mn-deficient plants remained the same until the 19th week of the experiment. The exception is the Chl *a*/Chl *b* ratio, which was decreased in Mn-deficient plants on the 6th week of the experiment and turned out to be lower than the control by 4.5%. At the 19th and 24th weeks of the experiment this ratio decreased by 11.6% and 12.6%, respectively. The first visual symptoms of chlorosis were noted on young upper needles starting from the 22nd week of the experiment and increased significantly at the 24th week. As a result, the Chl *a*, Chl *b* and Car contents were decreased by 37.3%, 28.3% and 23.1%, respectively, in Mn-deficient needles in comparison with the control. In addition, the Car/Chls ratio at the 24th week of the experiment was increased by 18.2% in Mn-deficient needles (Table 3).

Despite the clear signs of chlorosis caused by Mn deficiency at the 24th week of the experiment, we were unable to detect signs of oxidative stress. This is confirmed by the contents of MDA and 4-HNE being comparable in control and Mn-deficient plants, both in the needles and in the roots (Table 4).

### 3.4. Chlorophyll Fluorescence

Under control conditions, the primary photosynthetic processes in pine were efficient, as indicated by high F_v_/F_m_ values, high effective quantum yield of PSII, and the high coefficient of photochemical quenching (qP) (Table 5).

Mn deficiency caused a strong drop (38.2%) in the effective quantum yield of PSII (Y(II)) in comparison with the control. This Y(II) decrease was due to the strong increases in the quantum yield of non-regulated (Y(NO)) and regulated (Y(NPQ)) non-photochemical energy dissipation in PSII of 44.5% and 41.5%, respectively. At the same time, the maximum quantum yield of PSII photochemistry (F_v_/F_m_), was drastically reduced (by 24.8%) in comparison with the control, owing to fewer electrons passing through the photosystems (F_m_ was reduced by 27.3%) and as a result of increased photoinhibition of PSII (F_0_ was increased by 66.4%). qP increased by 10.8% and the electron transport rate of PSII (ETRII) was strongly reduced (by 38.0%) in Mn-deficient plants (Table 5).

### 3.5. Transcript Levels of Photosynthetic Genes and Genes Involved in Chlorophyll Biosynthesis

The transcripts of the *Lhcb2* and *Lhca* genes encoding the PSII light harvesting chlorophyll *a*/*b* binding protein and type 4 protein of light-harvesting complex (LHC) of PSI were 3.2- and 2.2-fold higher, respectively, than those in the control at the 24th week (Figure 4, Appendix A).

The transcripts of genes encoding the PSII core complex significantly increased in Mn-deficient plants at the end of the experiment compared with the corresponding control. Specifically, transcripts of the *psbA* and *psbD* genes encoding reaction center core proteins D1 and D2 were 1.5- and 2.1-folds higher than those in the corresponding control. The transcripts of the *psbC* and *psbB* genes encoding core antenna proteins CP43 and CP47, respectively, were 1.9- and 1.7-folds higher than those in the control (Figure 4).

The transcripts of the *psbO*, *psbP*, and *psbQ* genes encoding the corresponding extrinsic OEC proteins were 2.3-, 1.6- and 2.1-folds higher, respectively, than those in the control at the 24th week. The transcript of the *psbS* gene encoding the PsbS protein, which is involved in the development of nonphotochemical quenching [32], was 1.7-fold higher than that in the control. Increased transcript levels of the *psbO*, *psbQ* and *psbS* genes in Mn-deficient plants were also observed at the 6th week of the experiment (Figure 4, Appendix A).

Among the genes encoding the components of the Cyt*b_6_f* complex, the greatest increase in the transcript level was observed for the *petA* gene, encoding Cytochrome *f*, which was 2.7-fold higher than that in the control. The transcript of the *petD* gene, which encodes subunit 4 of the Cyt*b_6_f* complex, was 2.3-fold higher in Mn-deficient plants than that in the control. The least pronounced differences were found for the *petE* and *petC* genes encoding plastocyanin minor isoform and iron-sulfur subunit of the Cyt*b_6_f* complex, respectively. At the 24th week, their transcript levels in Mn-deficient plants exceeded the control by 1.7- and 1.4-folds, respectively. However, the transcript levels of these genes were increased in Mn-deficient plants at the 6th week of the experiment (Figure 4).

The transcript levels of *psaA* and *psaB* genes encoding the representative subunits of PSI (PsaA and PsaB) exceeded the control in Mn-deficient plants at the 6th and 24th weeks of the experiment. The maximal levels of their transcripts observed at the end of the experiment were 1.7- and 1.9-folds higher than those in the control (Figure 4).

The transcript of the *rbcL* gene encoding the large subunit of Rubisco was 2.8-fold higher in Mn-deficient than in the control plants at the end of the experiment. Additionally, in Mn-deficient plants, the transcript level of the *porA* gene encoding light-dependent NADPH-protochlorophyllide oxidoreductase (LPOR) exceeded the control by 1.6-fold. At the same time, the transcript levels of the *chlB*, *chlN* and *chlL* genes encoding subunit B, iron-sulfur ATP-binding protein and subunit N of light-independent protochlorophyllide reductase (DPOR), respectively, were 3.5-, 3.2- and 3.7-folds higher than those in the control (Figure 4).

Most of the studied genes were characterized by a gradual increase in transcript levels in Mn-deficient plants during the experiment, with maximum transcript levels at the 24th week. At the same time, for control plants, a transient increase in the transcript levels was noted at the 19th week, comparable to or even exceeding the maximum level of transcript levels in Mn-deficient plants (for *petE*, *porA* and *chlN* genes) (Figure 4). Apparently, this was due to the beginning of secondary needle development [33].

## 4. Discussion

### 4.1. Development of Mn Deficiency and Features of Scots Pine Seedling Growth 

The experimental system used made it possible to observe a gradient decrease in Mn content in the needles of plants grown on Mn-deficient solution to 0.05 µmol/g in young needles at the 24th week of the experiment (Figure 1B). In another experiment with Scots pine, lasting 27 weeks, the authors failed to achieve a decrease in the Mn content in young needles below 0.87 µmol/g. According to the authors, this could be related to the original amount of Mn supplied in the seed [19]. However, we found that the amount of Mn in the seeds was only sufficient to provide seedlings no more than 4 weeks. Even taking into account the variability in the original amount of Mn in the seeds, this is unlikely to be sufficient to provide older seedlings with Mn. For example, the maximum Mn content we observed in Scots pine seeds was 72.9 ± 3.0 nanomoles per individual seed [24], which is only 1.7-fold higher than that in the current experiment. This amount is comparable to the Mn amount per Mn-deficient 6-week-old seedling (Table 1).

A decrease in the Mn content in the needles of 4- and 6-week-old Mn-deficient plants from 0.61 to 0.34 µmol/g, respectively, did not lead to suppression of growth processes. In contrast, the growth of Mn-deficient plants during this period was enhanced due to the elongation of the main root and seedling needles (Table 2). We also did not observe cessation of the formation of lateral roots [5] or necrosis of root tips [2] typical for crops, even though the Mn content in the roots of Mn-deficient plants was drastically decreased (Figure 1A).

The minimum Mn contents noted earlier in the organs of 6-week-old Scots pine seedlings under the action of zinc [17,18] or copper [18,23] were several times higher than those in the needles and ten times higher than those in the roots of 6-week-old Mn-deficient plants. Consequently, disturbances in Mn uptake by pine seedlings observed under heavy metal stress are unlikely to cause growth retardation *per se*. At the same time, the Mn content in the needles of Mn-deficient 6-week-old plants was comparable to the Mn content in the needles of pine undergrowth (0.42 µmol/g) growing on soils with low availability of Mn [24]. Mature pine stands growing in these areas demonstrated a decrease in the yield and sowing qualities of seeds due to Mn deficiency, although basic growth processes were not disturbed [24].

As noted earlier, the lack of definitive information on the critical level of Mn in foliage from most conifer species [19] makes it very difficult to assess the requirements of Mn for Scots pine. According to Brække and Salih [21], a Mn content in current-year-needles of Scots pine greater than 0.27 µmol/g was defined as the optimum, while a Mn content lower than 0.18 µmol/g was defined as a strong deficiency. However, in our experiment a decrease in Mn content in the needles of Mn-deficient plants during the period of the 6th–12th weeks to 0.21 µmol/g (Figure 1B) was accompanied by a drastic decrease in plant growth compared to the control (Figure 2A,B). According to Ulrich and Hills [34], there are four ranges (zones), representing different physiological stress levels: optimum, pre-optimum (80–100% of maximum growth), deficiency (50–80% of maximum growth), and strong deficiency (< 50% of maximum growth). Thus, within 6 weeks, the Mn-deficient plants shifted from the optimum to the deficiency zone immediately, bypassing the pre-optimum zone. In our opinion, this is possible only when control plants are sufficiently supplied with Mn, providing maximal growth. This is also evidence that the lower limit of the optimum range for Mn content in Scots pine should be significantly higher than that indicated by Brække and Salih [21].

Starting from the 6th week of the experiment, the Mn content in the needles of control plants remained almost constant at the level of 6.1 µmol/g (Figure 1B). This is very close to the results obtained by Kavvadias and Miller [19] and within the optimal range of Mn content in one-year-old needles of Scots pine proposed by Oleksyn et al. [20]. Moreover, from northernmost Finland to south Poland the Mn content in Scots pine needle litter ranged from 68.1 to 4.7 µmol/g with a significant negative correlation with a mean annual temperature [1]. Based on these data, the minimum average Mn content in Scots pine needles should lie in the range of 2.3–3.1 µmol/g, whereas the newly shed needle litter mainly consists of older needles, containing 1.5–2-folds more Mn than one-year-old needles [14,24]. However, even in high-productive Scots pine stands there may be fluctuations in the Mn content in the needles during the growing season. For example, we found a decrease in Mn content in current-year needles during August by 2.0–2.7-folds, to a level of 2.0 µmol/g [15]. Therefore, such significant seasonal fluctuations in the Mn content seriously complicates the comparison of results obtained at different times of the year, while neither the causes nor physiological consequences of these fluctuations are currently known. However, based on available data, the lower limits of Mn content in current-year- or one-year-old needles sufficient for optimum growth of Scots pine should be in the range of 1.3–2.3 µmol/g. 

### 4.2. Physiological Processes in Scots Pine Seedlings under Mn Deficiency

It is known that strong Mn deficiency usually corresponds to visible deficiency symptoms [5,21]. According to Grey [11], the first reported symptoms associated with Mn deficiency in *Pinus radiata* were (1) the characteristic bronze color of the needles; (2) a marked reduction in height growth; and (3) lack of lateral branch development. Mn deficiency symptoms, as a bronzing of needle tips, were also observed on occasional plants of Scots pine with Mn content in young needles at 0.87 µmol/g [19]. In our experiment, despite Mn deficiency, which reduced plant growth at the 12th week, the content of photosynthetic pigments remained at a level comparable to the control until the 19th week (Table 3), although the Mn content in the needles was only 0.11 µmol/g. From the 22nd week of the experiment visual symptoms of chlorosis were observed on young needles, due to the low phloem mobility of Mn [2]. At the 24th week, the contents of photosynthetic pigments in Mn-deficient plants were significantly lower than those in the control (Table 3); however, we did not observe bronzing of needle tips. This was probably due to the fact that the irradiance in the growth chamber was lower than this when photoinhibition of photosynthesis occurs [35,36].

The dynamics of pigment contents in the needles of control plants reflect the development of photosynthetic organs. A gradient decrease in needle water content, which was more pronounced in control plants (Figure 2D), highlights the maturation of the needles. Scots pine is photophilous [31,37]; however, in the first few seasons of their life, seedlings can only be higher than the surrounding herbaceous plants for a short period of time, which causes a lack of light [31]. For this reason, the Chl *a* and Chl *b* contents in cotyledons (data not presented) are 2–3-fold higher than those in juvenile needles. The pigment content in the needles decreases with seedling growth; it helps to decrease light energy absorption and to therefore protect plants from photo-oxidative damage [38]. This is also confirmed by experiments with light of different spectral compositions, when high-energy light led to a decrease in the content of pigments in Scots pine needles [31]. However, the pigment contents in 24-week-old plants in the experiment remained two-fold higher than those in the current-year needles of Scots pine undergrowth in early August [15].

Among the main reasons for the development of chlorosis in Mn-deficient plants are disturbances in chlorophyll biosynthesis [5,39], chlorophyll degradation [2], changes in chloroplast ultrastructure [5,40] and changes in ion homeostasis [41]. However, we can exclude the lack of Mg, Fe or Cu as possible reasons for needle chlorosis as their content was not decreased in Mn-deficient plants (Figure 3B, Appendix A). In contrast, the transcript levels of genes encoding DPOR and LPOR substantially increased in the needles of Mn-deficient plants starting from the 6th week of the experiment (Figure 4). Consequently, the decrease in the pigment contents in the needles of Mn-deficient plants was due to other reasons.

It is known that in *Pinus thunbergii* the Chl *a*/Chl *b* ratio is highest in current needles and decreases slightly only as the needles age [33]. The Chl *a*/Chl *b* and the Car/Chls ratios are only slightly decreased in Mn-deficient needles (Table 3), which can be interpreted as indicating that the amount and type of pigments in the light-harvesting antennae serving the reaction centers were not significantly changed with Mn deficiency. It was shown that Mn-deficient chloroplasts from spinach leaves contained more Chl *b* than Chl *a* [42]. Thus the Chl *a*/Chl *b* ratio is an index of Mn deficiency. Furthermore, under Mn deficiency, no structural changes were found in the chloroplasts of different plants [35,40,42]. Henriques [40] proposed the explanation that the leaf reduces the number of chloroplasts according to the available amount of Mn in order to meet the Mn requirement of the remaining chloroplasts and, thus, assure their full functional competence.

Mn deficiency has detrimental effects on the photosynthetic apparatus owing to reduced photosynthetic electron transport and oxidative stress [8]. The increase in minimum Chl fluorescence (F_0_) and concomitant decrease in F_v_/F_m_ in Mn-deficient plants indicated profound defects in electron transfer within PSII (Table 5). In addition to F_v_/F_m_, Y(NO) was significantly increased in Mn-deficient plants indicating photoinhibition of PSII. Nevertheless, the increased level of Y(NPQ) in Mn-deficient plants indicated that some excess energy excitation was dissipated harmlessly as heat, which could prevent the generation of reactive oxygen species (ROS) and oxidative stress development (Table 4). Since ROS not only directly cause photodamage to PSII but also inhibit the repair of PSII through inhibiting protein synthesis [43], the absence of oxidative stress indicated that the repair could take place.

The development of oxidative stress under Mn deficiency is at least partly due to a decrease in the activity of Mn superoxide dismutase (MnSOD) [2,8]. A study of the green algae *Chlamydomonas* under Mn-deficient conditions showed a loss of MnSOD activity before that of PSII efficiency, suggesting a regulated intraorganellar supply of Mn to support PSII function in preference to MnSOD function in the mitochondria [4,44]. However, the activity of MnSOD in Scots pine needles corresponded to approximately 1 to 4% of the total SOD activity [45]. We believe that due to this unique feature of Scots pine a possible decrease in MnSOD activity under Mn deficiency does not pose a critical problem unlike most plant species.

### 4.3. Ion Homeostasis under Mn Deficiency

Despite the available evidence for secondary deficiency of P and Fe in Mn-deficient *Chlamydomonas* [44], ion homeostasis under Mn deficiency in higher plants has not been studied enough. However, it was shown that Mn deficiency led to a significant increase in Fe content in shoots of *Arabidopsis thaliana* [41,46] and in the leaves of mulberry (*Morus alba* L.) [47], while no changes were found in the Fe content in the needles of *Pinus banksiana* [22]. The shoots of *A. thaliana cmt1-1* (chloroplast manganese transporter1) mutants contained 30% more P than wild-type plants [46]. Wei Yang et al. [41] observed an approximately two-fold decrease in K and Zn contents in the roots of Mn-deficient *Arabidopsis*, while their contents in shoots were the same as those in the control. Under Mn deficiency, the contents of Zn and Cu increased in the leaves of mulberry [47], while no changes were found in the shoots of *Arabidopsis* [41]. We did not observe a decrease in the content of mineral nutrients in the roots of Mn-deficient seedlings during the experiment. In contrast, the K and especially Zn contents increased under Mn deficiency (Figure 3A, Appendix A). In the needles of Mn-deficient plants, there was also no decrease in the contents of any nutrients. The increased K content in the needles throughout the experiment, as the increased Mg and Fe contents in the needles at the end of the experiment were the most characteristic changes in Mn-deficient plants (Figure 3B, Appendix A). Thus, Mn deficiency *per se* does not provoke deficiencies of other nutrients in Scots pine.

Mn is a metal cofactor for approximately 6% of all known metalloenzymes, but only a few metalloenzymes are currently known to have an absolute and non-replaceable requirement for Mn to become catalytically active [4]. For the majority of Mn containing metalloenzymes, Mn is interchangeable with other divalent metal cations, such as Mg, albeit with lower efficiency [2,4,6]. Therefore, an increase in the Mg content in the needles of Mn-deficient plants may be associated with the need to maintain the activity of certain enzymes under critical Mn deficiency. Nevertheless, it should be noted that the Mg content in the needles of control plants already exceeds the Mn content by one order of magnitude (Figure 3B); therefore, such a significant increase in the Mg content in Mn-deficient plants is hardly necessary for Mn replacement in some enzymes. On the other hand, an increase in the Mg and Fe contents in the needles of Mn-deficient plants occurred against a decrease in the Chls contents. This should lead to the release of large amounts of Mg and Fe regardless of the reasons: reducing the number of chloroplasts [40] or disruption of chloroplasts. Thus, the reasons for the increase in the content of some nutrients under Mn deficiency remain unclear.

### 4.4. The Transcript Levels of Photosynthetic Genes under Mn Deficiency

In higher plants, a lumenal cap of three extrinsic proteins, PsbO, PsbP, and PsbQ, shields and protects the Mn_4_CaO_5_ cluster. This lumenal Mn protein complex forms the OEC of PSII [8,48]. The extrinsic protein PsbP, acts as a Mn carrier protein to introduce Mn into the OEC reaction center, where it subsequently stabilizes the Mn cluster in association with PsbO and PsbQ [2,48]. Among the 25 core PSII protein subunits, PsbO appears to be of prime importance in the photosynthetic water-splitting process and in OEC stabilization [49]. Mn deficiency alters the macro-organization of PSII, as demonstrated by a pronounced decrease in the abundance of the extrinsic proteins PsbP and PsbQ (but not PsbO) and D1 (PsbA) [4].

It was previously found that the majority of robustly Mn-regulated genes in *Arabidopsis* were down-regulated in response to Mn deficiency. Specifically, 105 transcripts and proteins related to photosynthesis, e.g., subunits of PSII and PSI, were strongly down-regulated [39]. Transcription and translation of chloroplast genes were also down-regulated in the *Arabidopsis cmt1-1* mutant: transcript levels of *psaA*, *psbA*, *psbB*, *psbC*, *psbD*, and *rbcL* were reduced to approximately 50% of wild type levels, whereas the levels of *petB* and *Lhcb2* transcripts were less affected. The accumulation of PSII subunits (D1, CP47, CP43, D2, PsbO, PsbP), PSI subunits (PsaA, PsaD), and RbcL was reduced to approximately 50% of wild type levels in *cmt1-1* [46]. Integrated transcriptomic and proteomic analyses of maize leaves showed that a considerable number of genes encoding proteins in the photosynthetic apparatus were only suppressed by Mn deficiency [36].

Surprisingly, we did not find a decrease in the transcript levels of any of the studied genes in Mn-deficient plants, with the exception of *psbD* at the 6th and 19th weeks of the experiment (Figure 4). In contrast to the control, Mn-deficient plants had an increased transcript abundance of genes encoding OEC, namely *psbO*, *psbP*, and *psbQ*, throughout the experiment. The transcript levels of chloroplast genes encoding other subunits of PSII, the Cyt*b_6_f* complex, PSI, the large subunit of Rubisco, and nuclear genes encoding LPOR and DPOR were also increased in Mn-deficient plants.

Needle expansion is accompanied by the active development of chloroplasts with *de novo* synthesis of chloroplast proteins, and chloroplast components are maintained in a functional state for more than two years. However an exceptionally high level of the *psbA* transcript suggests the rapid turnover of the Dl polypeptide [33]. It is known that the Mn cluster together with OEC proteins are disassembled and released in parallel with the damaged D1 under the PSII repair process [8,43]. Non-damaged protein components of PSII are recycled, but whether Mn ions are recycled during PSII repair remains unknown [4]. It has been speculated that Mn is recycled during PSII repair, and a secondary role for PsbP as a Mn storage protein during PSII repair has been proposed [8]. Thus, we can assume that an increase in the transcript levels of the *psbO*, *psbP*, *psbQ*, *psbA*, *psbB*, *psbC*, *psbD* genes in Mn-deficient plants is associated with PSII repair processes, and the increase in the transcript of the *psbS* gene is associated with the need for PsbS protein synthesis, which is associated with the proper dissipation of excess light energy via regulation of NPQ [32,50]. However, the reasons for the increase in the transcript levels of genes associated with LHCII, the Cyt*b_6_f* complex, LHCI, PSI and Rubisco in Mn-deficient plants remain unclear at present.

## 5. Conclusions

We found that the original amount of Mn in the seed was sufficient for the development of seedlings for 4 weeks, with a maximum for 6 weeks. During this period, the main root growth increased, which can contribute to root penetration into soil patches richer in Mn under natural conditions. When the Mn content in needles decreased below 0.34 µmol/g DW, plant growth was suppressed. At the same time, even at an extremely low Mn content (0.11 µmol/g) in the needles the content of photosynthetic pigments remained at a level comparable to the control. The development of Mn deficiency was not accompanied by deficiencies in other mineral nutrients, namely K, Mg, Ca, P, Fe, Zn and Cu, either in the roots or in the needles. Despite the pronounced photoinhibition of PSII, we did not observe the development of oxidative stress in the organs of seedlings. In contrast to those in herbaceous plants [36,39,46], the transcript levels of photosynthetic genes in Scots pine did not decrease under Mn deficiency. In contrast to the control, Mn-deficient plants had increased transcript levels of genes encoding OEC (namely *psbO*, *psbP*, and *psbQ*), other subunits of PSII (*Lhcb2*, *psbA*, *psbB*, *psbC*, *psbD*, *psbQ*), the Cyt*b_6_f* complex (*petA*, *petC*, *petD*, *petE*), PSI (*Lhca*, *psaA*, *psaB*), the large subunit of Rubisco (*rbcL*), LPOR (*porA*) and DPOR (*chlB*, *chlL*, *chlN*). Thus, the patterns of transcript levels of genes encoding Mn-dependent enzymes, such as the other enzymes of PSII and PSI, which do not require Mn directly, and other enzymes supporting photosynthesis, are very similar under Mn deficiency. The reason for this relationship is unknown but suggests a common pathway for regulating the expression of these genes.

We believe that the high requirement of Scots pine for Mn, as well as the high natural variability in the Mn content in needles [1], is associated with the maintenance of the functionality of the photosynthetic apparatus at different light intensities. It is known that changes in light intensity and Mn status can interfere with the functionality of the photosynthetic apparatus [36]. The loss of chloroplast manganese transporter1 (CMT1) in *Arabidopsis* results in severely reduced photosynthetic activity and markedly enhanced susceptibility to high light. However, the diminished PSII function in the *cmt1-1* mutant caused by Mn depletion in chloroplasts can be alleviated by increasing Mn availability to the plant [46]. Previously we showed that the Mn content in the needles of undergrowth growing under the forest canopy was 1.6–2.8-folds lower than that in the needles of mature trees in the same areas [24]. Moreover, a clear trend of increasing Mn content in Scots pine needles from south Poland to northernmost Finland with a significant negative correlation with a mean annual temperature [1] indicates the involvement of Mn in protecting the photosynthetic apparatus under conditions of high light and low temperatures in winter.

## Figures and Tables

**Figure 1 cells-11-03814-f001:**
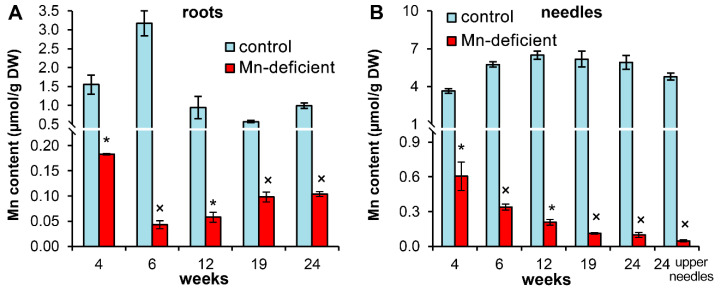
The Mn content in the organs of Scots pine seedlings: roots (**A**); and needles (**B**). The mean values ± SEs are given (*n* = 6). Asterisks (*, for Student’s *t* test) or multiplication signs (×, for Mann–Whitney rank sum test) denote significant differences at *p* < 0.05 between control and Mn-deficient plants at each separate time point.

**Figure 2 cells-11-03814-f002:**
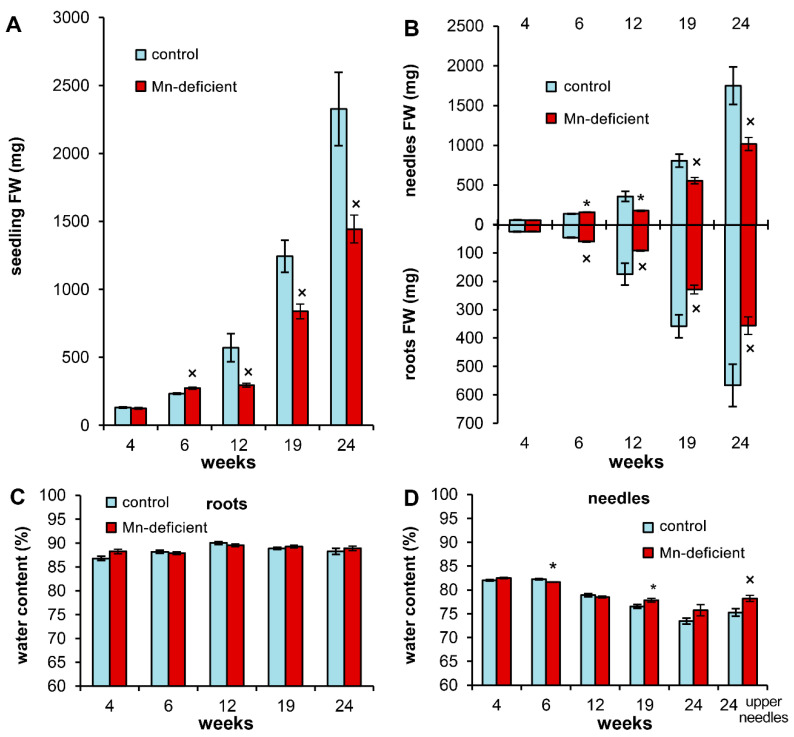
The fresh weight (**A**,**B**) and water content (**C**,**D**) in the organs of Scots pine seedlings during the experiment. Asterisks (*, for Student’s *t* test) or multiplication signs (×, for Mann–Whitney rank sum test) denote significant differences at *p* < 0.05 between control and Mn-deficient plants at each separate time point.

**Figure 3 cells-11-03814-f003:**
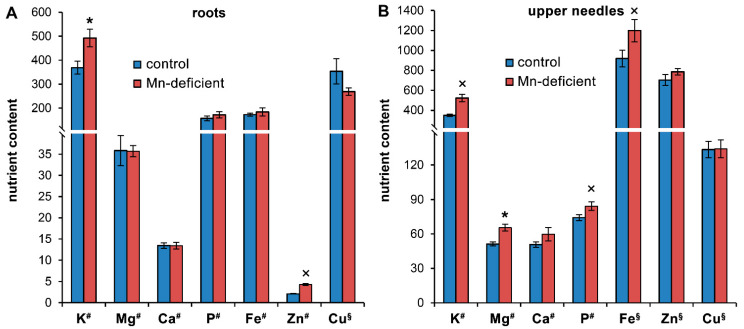
The contents of nutrients in the organs of Scots pine seedlings at the 24th week of the experiment: roots (**A**); upper needles (**B**). The mean values ± SEs are given (*n* = 6). The mean element contents in µmol/g DW (#) or nmol/g DW (§). Asterisks (*, for Student’s *t* test) or multiplication signs (×, for Mann–Whitney rank sum test) denote significant differences at *p* < 0.05 between control and Mn-deficient plants.

**Figure 4 cells-11-03814-f004:**
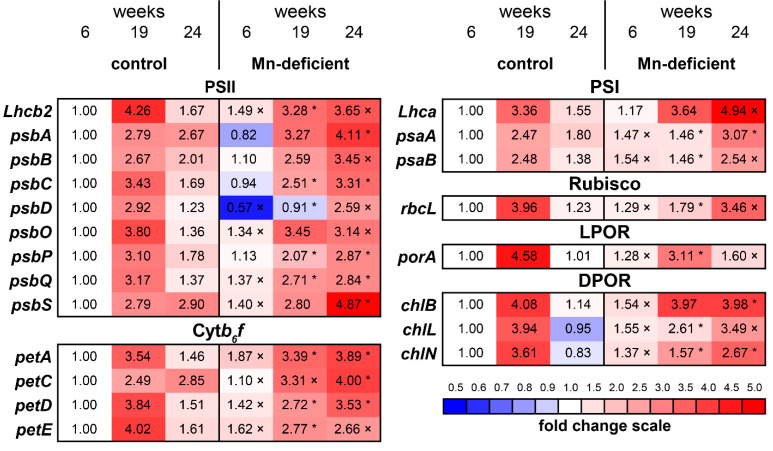
The transcript levels of photosynthetic genes (*Lhcb2*, *psbA*, *psbB*, *psbC*, *psbD*, *psbO*, *psbP*, *psbQ*, *psbS*, *petA*, *petC*, *petD*, *petE*, *Lhca*, *psaA*, *psaB* and *rbcL*) and genes involved in chlorophyll biosynthesis (*porA*, *chlB*, *chlL*, *chlN*) in the needles of Scots pine seedlings. Asterisks (*, for Student’s *t* test) or multiplication signs (×, for Mann–Whitney rank sum test) denote significant differences at *p* < 0.05 between control and Mn-deficient plants at each separate time point.

**Table 1 cells-11-03814-t001:** The Mn amounts per seedling at the 4th and 6th weeks of the experiment.

Variant	4th Week	6th Week
Control	113.7 ± 12.9	228.3 ± 14.3
Mn-deficient	53.9 ± 2.8 *	70.3 ± 2.3 ^×^

Pairwise comparisons of the means were performed between control and Mn-deficient plants at each separate time point using Student’s *t* test for normally distributed data or the Mann–Whitney rank sum test when the *t* test was not applicable. Asterisks (*, for Student’s *t* test) or multiplication signs (^×^, for Mann–Whitney rank sum test) denote significant differences at *p* < 0.05.

**Table 2 cells-11-03814-t002:** Growth parameters of Scots pine seedlings at the 4th and 6th weeks of the experiment.

Parameter	Control		Mn-Deficient	
	4th Week	6th Week	4th Week	6th Week
Primary root length, mm	163.8 ± 6.0	222.3 ± 3.3	157.1 ± 8.4	244.4 ± 4.7 ^×^
The distance from the tip of the main root to the first lateral root, mm	64.4 ± 3.4	61.8 ± 1.9	67.1 ± 6.0	65.6 ± 2.1 ^×^
The number of first-order lateral roots, pieces	35.2 ± 2.2	53.6 ± 1.3	28.1 ± 4.3	50.8 ± 1.6
Average hypocotyls length, mm	30.1 ± 0.7	29.4 ± 0.3	28.3 ± 1.5	30.4 ± 0.5
Average needle length, mm	28.3 ± 0.7	33.5 ± 0.3	27.6 ± 1.0	35.1 ± 0.5 ^×^
Number of needles, pieces	19.4 ± 0.6	34.4 ± 0.5	17.9 ± 0.8	35.9 ± 0.8

Pairwise comparisons of the means were performed between control and Mn-deficient plants at each separate time point using the Mann–Whitney rank sum test when the *t* test was not applicable. Multiplication signs (^×^) denote significant differences at *p* < 0.05.

**Table 3 cells-11-03814-t003:** Content of photosynthetic pigments.

Parameter	Control			Mn-Deficient		
	6th Week	19th Week	24th Week	6th Week	19th Week	24th Week
Chl *a*, mg/g DW	7.09 ± 0.20	7.22 ± 0.58	4.99 ± 0.31	6.51 ± 0.19	6.93 ± 0.29	3.13 ± 0.12 *
Chl *b*, mg/g DW	2.92 ± 0.09	2.97 ± 0.24	2.09 ± 0.15	2.80 ± 0.08	3.23 ± 0.14	1.50 ± 0.02 ^×^
Carotenoids, mg/g DW	1.12 ± 0.03	1.06 ± 0.08	0.78 ± 0.04	1.06 ± 0.03	1.11 ± 0.04	0.60 ± 0.01 *
Chl a/Chl b	2.43 ± 0.02	2.43 ± 0.01	2.39 ± 0.03	2.32 ± 0.02 *	2.15 ± 0.04 ^×^	2.09 ± 0.09 ^×^
Car/Chls	0.112 ± 0.001	0.104 ± 0.003	0.110 ± 0.002	0.114 ± 0.002	0.109 ± 0.002	0.130 ± 0.005 *

Pairwise comparisons of the means were performed between control and Mn-deficient plants at each separate time point using Student’s *t* test for normally distributed data or the Mann–Whitney rank sum test when the *t* test was not applicable. Asterisks (*, for Student’s *t* test) or multiplication signs (^×^, for Mann–Whitney rank sum test) denote significant differences at *p* < 0.05.

**Table 4 cells-11-03814-t004:** Content of lipid peroxidation products in Scots pine at the 24th week of the experiment.

**Parameter**	**Roots**		**Needles**	
	**Control**	**Mn-Deficient**	**Control**	**Mn-Deficient**
MDA, nmol/g DW	81.6 ± 7.4	67.2 ± 3.4	53.9 ± 2.1	57.7 ± 2.3
4-HNE, nmol/g DW	183.9 ± 25.5	188.6 ± 9.0	933.3 ± 114.3	1093.8 ± 127.6

**Table 5 cells-11-03814-t005:** Chlorophyll fluorescence parameters of needles at the 24th week of the experiment.

Parameter	Control	Mn-Deficient
F_v_/F_m_	0.836 ± 0.004	0.629 ± 0.008 *
F_0_	175.3 ± 11.3	291.7 ± 36.0 *
F_m_	1075.0 ± 82.5	781.7 ± 92.9 *
Y(II)	0.531 ± 0.013	0.328 ± 0.014 *
Y(NO)	0.281 ± 0.011	0.406 ± 0.011 *
Y(NPQ)	0.188 ± 0.016	0.266 ± 0.013 *
qP	0.715 ± 0.016	0.792 ± 0.017 *
ETRII	27.7 ± 0.6	17.2 ± 0.7 *

Pairwise comparisons of the means were performed between control and Mn-deficient plants using Student’s *t* test for normally distributed data. Asterisks (*) denote significant differences at *p* < 0.05.

## Data Availability

The datasets generated and/or analyzed during the current study are available from the corresponding author on reasonable request.

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
