# Peer review of "Manganese Deficiency Suppresses Growth and Photosynthetic Processes but Causes an Increase in the Expression of Photosynthetic Genes in Scots Pine Seedlings"

_cells, 2022, doi:10.3390/cells11233814_

Round 1

Reviewer 1 Report

In this paper, the authors carried out the functional study on the influence of Mn deficiency in a Coniferous species (Pinus sylvestris L.) to investigate the response of growth, ion homeostasis and content of photosynthetic pigments, transcript levels of photosynthetic genes and genes involved in chlorophyll biosynthesis. Overall results are fruitful as Mn is an important micronutrient for plant growth and development and sustains metabolic roles.

In general, the paper is well-written and as such, it is very easy to read and understand. I also see no severe methodological flaws but several sections need to be improved before the paper can be accepted.
There are some points or recommendations for the authors which should be considered for this article.

For example:
Methodology
L169: Lichtenthaler formula => (mention this formula in methodology section)
Results
3.1. The Mn content in the nutrient solutions, seeds and seedling organs
L241-244: => (the values should be mentioned in the form of table as well, there is no table for this in text and supplementary file for this)
Discussion
L403-404: This amount is comparable to the Mn amount per Mn-deficient 6-week-old seedling (see Results). => Please explain this in discussion part.

467-468: At the 24 th week, the contents of photosynthetic pigments in Mn-deficient plants were significantly lower than those in the control => (mention the figure or table here).

Minor points:
L95: [19] found => mention the citation as Kavvadias et al. [19]
L96: while [21] =>  mention the citation as Brække and Salih [21]
L97: [11] => Grey [11]
L98: and [22] => Morrison and Armson [22]
Also correct the citations by including the author name in the text for the lines L175,186, 198, 199, 200, 424, 429, 437,440, 441, 458.

L533: Wei Yang [41] => Wei Yang et al. [41]
L549: Therefore => (Add comma)
L551: Nevertheless => (Add comma)

Also, check the formatting of the references list. In particular:
L680:  EURASIAN SOIL SCIENCE C/C OF POCHVOVEDENIE =>lowercase font

L704, L755: Pinus Sylvestris L => italics

L727: Scientific reports=> Scientific Reports

L757: CHLOROPLAST MANGANESE TRANSPORTER1=> lowercase font

L759: Morus Alba=> italics
I hope the authors find these suggestions useful to improve the paper.

Author Response

We are grateful to the Reviewer for the appreciation of the manuscript and clear suggestions for its improvement.

Question 1. L169: Lichtenthaler formula => (mention this formula in methodology section).

Answer 1. The necessary formulas were added to the manuscript.

Question 2. L241-244: => (the values should be mentioned in the form of table as well, there is no table for this in text and supplementary file for this)

Answer 2. As per your suggestion, instead of describing the results in the text, a table has been added to the manuscript.

Question 3. L403-404: This amount is comparable to the Mn amount per Mn-deficient 6-week-old seedling (see Results). => Please explain this in discussion part.

Answer 3. The link to Table 1 was added.

Question 4. 467-468: At the 24 th week, the contents of photosynthetic pigments in Mn-deficient plants were significantly lower than those in the control => (mention the figure or table here).

 Answer 4. The link to Table 3 was added.

Question 5. L95: [19] found => mention the citation as Kavvadias et al. [19]
Answer 5. According to your suggestion, the link to reference 19 was corrected.

Question 6. L96: while [21] =>  mention the citation as Brække and Salih [21]
Answer 6. According to your suggestion, the link to reference 21 was corrected.

Question 7. L97: [11] => Grey [11]
Answer 7. According to your suggestion, the link to reference 11 was corrected.

Question 8. L98: and [22] => Morrison and Armson [22]
Answer 8. According to your suggestion, the link to reference 22 was corrected.

Question 9. Also correct the citations by including the author name in the text for the lines L175,186, 198, 199, 200, 424, 429, 437,440, 441, 458.

Answer 9. The citations were corrected.

Question 10. L533: Wei Yang [41] => Wei Yang et al. [41]
Answer 10. The citation was corrected.

Question 11. L549: Therefore => (Add comma)
Answer 11. The comma was added.

Question 12. L551: Nevertheless => (Add comma)
Answer 12. The comma was added.

Question 13. Also, check the formatting of the references list. In particular:
L680:  EURASIAN SOIL SCIENCE C/C OF POCHVOVEDENIE =>lowercase font

Answer 13. This inaccuracy was fixed.

Question 14. L704, L755: Pinus Sylvestris L => italics

Answer 14. This inaccuracy was fixed.

Question 15. L727: Scientific reports=> Scientific Reports

Answer 15. This inaccuracy was fixed.

Question 16. L757: CHLOROPLAST MANGANESE TRANSPORTER1=> lowercase font

Answer 16. This is not a mistake. This is an original title of the manuscript.

Question 17. L759: Morus Alba=> italics
Answer 17. This inaccuracy was fixed.

Reviewer 2 Report

Mn nutritional requirement for the plant growth, metabolism and it is an essential cofactor for may enzymes particularly oxygen-evolving complex (OEC) of photosystem II.  It is a very dynamic process depending on the growth phase of the plant, I do not understand the rationale behind this study? If we anlyse these genes expression in Mn depleted condition after 15 days, the gene expression scenario will be stabilized or altered, thus it’s dynamic process? It is advised to write 1-2 sentence about the nature of these genes and its context upon role of Mn as a cofactor towards chlorophyll and function fo OEC –PSII in abstract, intro and final conclusion. Otherwise the Biological impact of Mn associated gene expression and its relevance in PS-II remains incomplete.

Major concern:

1.During Mn deficiency, how the enzymes behave in the absence of Mn cofactor?

2. DO the author can give an account requirement of Mn as essential or optional cofactor for the enzymes involved in chlorophyll synthesis, OEC function of PSII? Overall photosysnthic activity?

3. It was shown hear that during Mn Deficiency the chlorophyll content decreased however, on the contrast few genes of OEC level expressed more? Why this contrary? Is this being to compensate the function?   What could be the physiological scenario in this context?

4. Does the author made any effort upon supplementing the physically relevant Mn will it reached a stable expression?

5. By knowing few genes are overexpressed? In what way it could help to derive info or crop production strategy in Mn deficient soil conditions.  What is conclusion of the study?

6. Can the results suggests an alternate Co-factor for MN?

Author Response

We are grateful to the Reviewer for the comments and suggestions for the improvement of the manuscript.

Question. Mn nutritional requirement for the plant growth, metabolism and it is an essential cofactor for may enzymes particularly oxygen-evolving complex (OEC) of photosystem II.  It is a very dynamic process depending on the growth phase of the plant, I do not understand the rationale behind this study? If we anlyse these genes expression in Mn depleted condition after 15 days, the gene expression scenario will be stabilized or altered, thus it’s dynamic process? It is advised to write 1-2 sentence about the nature of these genes and its context upon role of Mn as a cofactor towards chlorophyll and function fo OEC –PSII in abstract, intro and final conclusion. Otherwise the Biological impact of Mn associated gene expression and its relevance in PS-II remains incomplete.

Answer. Among the proteins involved in photosynthesis, the first group of proteins (PsbP, PsbQ, PsbO, D1 and CP43) forms apo-OEC, which is directly associated with the Mn cluster. The proteins of another group do not require Mn directly but are involved in the functioning of the photosynthetic apparatus. It was important to find differences in the expression of genes encoding these distinct protein groups under Mn deficiency. In particular, it was not clear whether Mn deficiency would affect the level of transcripts of genes of Mn-dependent enzymes only or cause massive changes in the levels of transcripts of genes encoding proteins that do not require Mn directly. The necessary clarifications were added to the manuscript.

Major concern:

Question 1. During Mn deficiency, how the enzymes behave in the absence of Mn cofactor?

Answer 1. Under Mn deficiency, enzymes either lose their activity completely (absolute dependence on Mn), or their activity is significantly reduced. For some enzymes, the replacement of manganese with another divalent metal, in most cases with magnesium, was found, while the activity of the enzyme decreased (Schmidt and Hustead, 2019). Partially this was discussed in section 4.3 of the manuscript.

Question 2. DO the author can give an account requirement of Mn as essential or optional cofactor for the enzymes involved in chlorophyll synthesis, OEC function of PSII? Overall photosysnthic activity?

Answer 2. Mn has an indispensable role in the oxygen-evolving complex of PSII, which catalyses the oxidation of water to protons and molecular oxygen. Insufficient Mn supply leads to decreased oxygen evolution and, hence, to lower rates of photosynthesis and decreased plant growth. Other enzymes of PSI, PSII, the Cytb6f complex and LHC, such as the enzymes involved in chlorophyll biosynthesis, do not require Mn directly. However, our study showed that the expression of the genes encoding these proteins increased under Mn deficiency.

Question 3. It was shown hear that during Mn Deficiency the chlorophyll content decreased however, on the contrast few genes of OEC level expressed more? Why this contrary? Is this being to compensate the function?   What could be the physiological scenario in this context?

Answer 3. The development of chlorosis and a decrease in the content of photosynthetic pigments were not caused by deficiencies in other nutrients, particularly Mg. Thus, the decrease in chlorophyll content may be associated with a possible decrease in the number of photosystem reaction centers under Mn deficiency. This protective mechanism could prevent excessive absorption of light energy. On the other hand, an increase in the transcription of genes is not always accompanied by an increase in the content of the corresponding protein or the product of protein activity.

Question 4. Does the author made any effort upon supplementing the physically relevant Mn will it reached a stable expression?

Answer 4. At this stage of research, we set the task to identify the minimal Mn content in the organs of Scots pine seedlings, initiating disturbances of growth during Mn deficiency. The processes of plant recovery from Mn deficiency were not included in the objectives of this study but are planned in our future work.

Question 5. By knowing few genes are overexpressed? In what way it could help to derive info or crop production strategy in Mn deficient soil conditions.  What is conclusion of the study?

Answer 5. In our opinion, Mn deficiency in the soil cannot be effectively compensated for by any molecular genetic method. The simplest and most efficient techniques are soil fertilization or foliar Mn application. The aim of our study was to reveal the physiological and molecular consequences of Mn deficiency in Scots pine – the most widespread species of the Pinaceae family and an important tree species in Northern Eurasia.

Question 6. Can the results suggests an alternate Co-factor for MN?

Answer 6. It is known that for the majority of Mn containing metalloenzymes, Mn is interchangeable with other divalent metal cation (Schmidt and Husted, 2019). However, this question was outside the scope of our study.